# A decrease in rockfall probability under climate change conditions in Germany

Katrin M. Nissen[1], Martina Wilde[2,3], Thomas M. Kreuzer[3], Annika Wohlers[2], Bodo Damm[2], and Uwe Ulbrich[1]

[1]Institute for Meteorology, Freie Universität Berlin, Berlin, Germany
[2]Institute for Applied Physical Geography, University of Vechta, Vechta, Germany
[3]Institute for Geography and Geology, University of Würzburg, Würzburg, Germany

**Correspondence:** Katrin Nissen (katrin.nissen@met.fu-berlin.de)

**Abstract.** The effect of climate change on rockfalls in the German low mountain regions is investigated following two different approaches. The first approach uses a logistic regression model that describes the combined effect of precipitation, freeze-thaw cycles and fissure water on rockfall probability. The climate change signal for past decades is analysed by applying the model to meteorological observations. The possible effect of climate change until the end of the century is explored by applying the statistical model to the output of a multi-model ensemble of 23 regional climate scenario simulations. It is found that the number of days per year exhibiting an above-average probability for rockfalls has been mostly decreasing during the last decades. Statistical significance is, however, present only at few sites. A robust and statistically significant decrease can be seen in the RCP8.5 climate scenario simulations for Germany and neighbouring regions, locally falling below -10% when comparing the last 30 years of the 20th century to the last 30 years of the 21st century. The most important factor determining the projected decrease in rockfall probability is a reduction in the number of freeze-thaw cycles expected under future climate conditions.

For the second approach four large-scale meteorological patterns that are associated with enhanced rockfall probability are identified from reanalysis data. The frequency of all four patterns exhibits a seasonal cycle that maximizes in the cold half of the year (winter/spring). Trends in the number of days that can be assigned to these patterns are determined both in meteorological reanalysis data and in climate simulations. In the reanalysis no statistically significant trend is found. For the future scenario simulations all climate models show a statistically significant decrease in the number of rockfall promoting weather situations.

## 1 Introduction

The impact of climate change on the occurrence of landslides is an important research topic as landslides are responsible for numerous fatalities and cause high economic costs each year (Kirschbaum et al., 2015; Klose et al., 2015; Gariano and Guzzetti, 2016; Froude and Petley, 2018; Haque et al., 2019). Therefore, it is essential to investigate how meteorological changes will influence the occurrence of landslides in the future, to provide important information for planning and mitigation strategies (Schlögl and Matulla, 2018). The term landslide is general and encompasses several gravitational mass movement processes such as falls, flows, or slides and it is necessary to specify the types of mass movement and to analyse them separately (Varnes,

1978; Hungr et al., 2014). In this study, the focus lies on rockfall processes, which are defined as the detachment of rock from steep slopes or cliffs descends by falling, rolling, or bouncing. The main triggers are precipitation, snowmelt, and freeze-thaw cycles (Varnes, 1978; Cruden and Varnes, 1996; Wieczorek, 1996; Highland and Bobrowsky, 2008; Hungr et al., 2014).

In several reports, the Intergovernmental Panel on Climate Change (IPCC) states that in some regions it is highly likely that changes in heavy precipitation will have an effect on rockfall occurrence and that the frequency of rockfalls increases mainly due to permafrost degradation (IPCC, 2007, 2012, 2013, 2021). Additionally, numerous studies on the effects of climate change on rockfall predict that the frequency of rockfall will increase (Allen and Huggel, 2013; Ravanel and Deline, 2015) with changes varying between seasons (Stoffel et al., 2014). However, most of these studies were conducted for high mountain ranges, where permafrost plays a key role (Gruber and Haeberli, 2007; Ravanel and Deline, 2011, 2015; Stoffel et al., 2014). In contrast, studies focusing on lower mountain ranges came to different conclusions, emphasising that a general assumption on the impact of climate change on rockfalls is not realistic (Dehn and Buma, 1999; Collison et al., 2000; Sass and Oberlechner, 2012). For example, Sass and Oberlechner (2012) analysed over 250 rockfall events located in areas without permafrost in Austria, finding no indication of increased rockfall occurrence due to climate change. Mainieri et al. (2022) investigated the influence of global warming on rockfalls at low elevation regions in the French Alps. They applied multiple regression models with a rockfall inventory, based on tree ring analysis and investigated the effect of changes in climatic conditions based on reanalysis data. They concluded that for their sites an increase of rockfall occurrence related to climate change is not to be expected.

There are very few studies on the impact of climate change on rockfall occurrence that use climate change scenarios. Studies on climate change for other mass movement types include, for example, Jemec Auflič et al. (2021). These authors analysed the impact of climate change on shallow and deep-seated landslide between 1961 and 2070 in Slovenia using six different climate models for estimating the effect of the representative Concentration Pathway (RCP) climate scenario 4.5, counting precipitation threshold exceedances and investigating the size of the affected area. Their results indicate that more frequent extreme rainfall events may lead to an increase in landslide occurrence.

The literature cited here illustrates that the triggers and thresholds are site and process specific. The present study is the first that investigates rockfall occurrences under climate change conditions in lower mountain ranges in Germany. Results of two independent approaches are presented. For the first approach, the logistic regression model described by Nissen et al. (2022), that was trained using meteorological observations and a rockfall data set with events observed in Germany, is applied to meteorological timeseries data. For the second approach large-scale atmospheric patterns associated with an enhanced probability for rockfall initialisation are identified and changes in the frequency of these patterns are determined. In contrast to Luque-Espinar et al. (2017) who investigated links between large-scale atmospheric phenomena (e.g. ENSO, NAO and the QBO) to rockfall and landslide events in Spain based on spectral analysis, we identify relevant patterns in the 500 hPa geopotential height field over Europe with an approach based on cluster analysis. This type of approach was also used by Messeri et al. (2015) to develop a risk index for landslides in Italy.

In the present study the climate change signal is investigated for two different time periods: a) for the last decades using meteorological observations or reanalysis data and b) for future periods using a multi-model ensemble of climate scenario simulations.

## 2 Data

### 2.1 Rockfall data

The present study uses historical rockfall data that are extracted from a landslide database of Germany (Rupp and Damm, 2020; Damm and Klose, 2015). The rockfall data are gathered from scientific publications, governmental reports, police reports, civil protection reports, newspapers, field data collections, and GIS as well as web analyses. The landslide database of Germany currently contains about 7500 mass movement events of different types. The majority of events listed occurred in the last 200 years, the oldest event can be dated back to 1137. The data base is not comprehensive and the number of included events increase with time due to the fact that data on landslide observations became more readily available in recent years. Within the database, 670 rockfall events have been registered. In this regard, the generic term "rockfall" includes a variety of geo-morphological processes (e.g., rockfall, rock topple, debris fall, debris topple), which are characterised by a rapid gravitational downslope fall of debris or rocks, variable in particle size and volume (Varnes, 1978; Selby, 1993; Cruden and Varnes, 1996; Dorren, 2003). The rockfalls within the landslide database include information on the date and precise location of the event. For 343 events the exact day is known. From these, the 213 events that occurred after 1 January 1979 (start of the weather type classification) were included in the analysis. The extracted dataset is complemented by data provided and collected by the German railway company Deutsche Bahn (DB). The DB dataset covers the years 2015-2020 and includes 55 rockfall events. The data have been collected by the technical staff of the Deutsche Bahn. As we found no evidence for discrepancies, we included these data into our study, being aware of the risk that the level of expertise on which the classification is based might be limited and that this part of the rockfall data might inherit some delay considering the event occurrence, as the railway data set lists the day of event detection.

### 2.2 Meteorological data

The two gridded observational data sets that were used to fit the logistic regression model in Nissen et al. (2022) are used again in this study to estimate trends in climate related rockfall probability during the last decades. Daily minimum and maximum temperatures are taken from the E-OBS dataset (Cornes et al., 2018), while daily precipitation at the rockfall sites is extracted from the REGNIE data set (Rauthe et al., 2013). Both datasets are constructed from station observations interpolated to a regular grid considering orographical conditions. The horizontal resolutions of this grid is $0.1° \times 0.1°$ for E-OBS and 1 km$^2$ for REGNIE. For consistency, daily minimum and maximum from the E-OBS dataset are also used to remove potential biases from climate simulations prior to the identification of freeze-thaw cycles in these simulations.

The ERA5 data set is the basis for a customized classification of large-scale weather patterns. ERA5 is the fifth generation ECMWF atmospheric reanalysis of the global climate (Hersbach et al., 2020). For the present analysis daily data covering the period 1979-2020 was used. At the time of writing, data for the pre-satellite era 1950-1978 was still categorized as preliminary and therefore excluded from the analysis. The original data was smoothed by interpolating it to a regular $1° \times 1°$ grid.

### 2.3 Climate scenario simulations

Future climate conditions (2071-2100) are analysed from regional climate scenario simulations conducted under the EURO-CORDEX initiative (Jacob et al., 2014) by comparing the scenario period with data produced with the same models under present day greenhouse gas forcing (1971-2000). We selected the simulations with the highest available horizontal resolution of $0.11° \times 0.11°$ corresponding to approximately 12 km $\times$ 12 km. The regional simulations are driven by the output of global climate models from the CMIP5 initiative (Taylor et al., 2012). In order to capture the upper range of the potential change, the emission pathway RCP8.5 scenario was chosen. The analysis includes the model simulations accessible from the FREVA XCES data server (Kadow et al., 2021) set up for the ClimXtreme project (www.climxtreme.net). Table 1 lists these simulations. If more than one ensemble member was available only the first was used for the analysis in order ensure equal weighting of the models. In total 23 combinations of global and regional models were included in the analysis.

### 2.4 Definition of subregions

In the following we will sometimes refer to specific subregions. The Central European (CE) subregion is located north of the Alps and includes Germany and some neighbouring areas (2°E-24°E,47°N-56°N). Its extent is depicted in Fig. 2. The greater European subregion (GE) extends from 15°W to 30°E and from 35°N to 70°N (see Fig. 5). The smaller European subregion (SE) used for a sensitivity test covers the region 0°E-20°E 40°N-60°N (not shown).

## 3 Methods

### 3.1 Logistic regression

In Nissen et al. (2022) a logistic regression model was developed that describes the probability for rockfalls as a function of meteorological predictors:

$$p = 1 / \left[ 1 + \exp(+10.48 + 2.969 \times 10^{-3} \times \mathrm{precip_{lperc}} + 1.413 \times 10^{-2} \times D_{\mathrm{perc}} - \underbrace{0.435}_{\text{if ftc=TRUE}} - 4.053 \times 10^{-4} \times \mathrm{precip_{lperc}} \times D_{\mathrm{perc}}) \right].$$

(1)

where $\mathrm{precip_{lperc}}$ is daily precipitation expressed as local percentiles, $D_{\mathrm{perc}}$ is the percentile of a fissure water proxy and ftc denotes if a freeze-thaw cycle occurred in the previous 9 days. This 9-day period was identified as optimal in Nissen et al. (2022). The fissure water proxy $D$ is defined as the difference between precipitation accumulated over 5 days and the potential evapotranspiration (PET) during this period. PET is determined using the method proposed by Hargreaves (1994) in

**Table 1.** Climate scenario simulations analysed for this study.

| Driving Model | Regional Model | Reference |
|---|---|---|
| CNRM-CERFACS-CNRM-CM5 | SMHI-RCA4-v1 | Kupiainen et al. (2015) |
| ICHEC-EC-EARTH | SMHI-RCA4-v1 | " |
| IPSL-IPSL-CM5A-MR | SMHI-RCA4-v1 | " |
| MOHC-HADGEM2-ES | SMHI-RCA4-v1 | " |
| MPI-M-MPI-ESM-LR | SMHI-RCA4-v1 | " |
| ICHEC-EC-EARTH | DMI-HIRHAM5-v1 | Christensen et al. (1998) |
| NCC-NORESM1-M | DMI-HIRHAM5-v1 | " |
| IPSL-IPSL-CM5A-MR | IPSL-INERIS-WRF331F-v1 | Skamarock et al. (2008) |
| CNRM-CERFACS-CNRM-CM5 | CLMcom-CCLM4-8-17-v1 | Rockel et al. (2008) |
| ICHEC-EC-EARTH | CLMcom-CCLM4-8-17-v1 | " |
| MOHC-HADGEM2-ES | CLMcom-CCLM4-8-17-v1 | " |
| MPI-M-MPI-ESM-LR | CLMcom-CCLM4-8-17-v1 | " |
| MIROC-MIROC5 | CLMcom-CCLM4-8-17-v1 | " |
| CCCma-CanESM2 | CLMcom-CCLM4-8-17-v1 | " |
| MPI-M-MPI-ESM-LR | MPI-CSC-REMO2009-v1 | Jacob et al. (2012) |
| CNRM-CERFACS-CNRM-CM5 | KNMI-RACMO22E-v2 | Van Meijgaard et al. (2012) |
| MOHC-HADGEM2-ES | KNMI-RACMO22E-v2 | " |
| ICHEC-EC-EARTH | KNMI-RACMO22E-v1 | " |
| IPSL-IPSL-CM5A-MR | KNMI-RACMO22E-v1 | " |
| MPI-M-MPI-ESM-LR | KNMI-RACMO22E-v1 | " |
| NCC-NORESM1-M | KNMI-RACMO22E-v1 | " |
| MPI-M-MPI-ESM-LR | GERICS-REMO2015-v1 | Jacob et al. (2012) |
| NCC-NORESM1-M | GERICS-REMO2015-v1 | " |

the version modified by Droogers and Allen (2002). As input parameters it needs extraterrestrial radiation (which depends on latitude and day of the year), the period mean of maximum and minimum daily temperatures, as well as mean precipitation over the accumulation period (as a proxy for cloudiness). A freeze-thaw cycle is defined as the transition from a daily minimum temperature below $0°C$ to a daily maximum temperature higher than $0°C$. Thus, the three variables needed for the model are daily minimum and maximum temperatures (Tmax and Tmin) as well as daily precipitation.

First, the logistic regression model is used to analyse trends in the observational time period (1950-2020) using observational data (E-OBS and REGNIE) at the rockfall sites as input. Trends in the annual number of days per year with $p > p_{clim}$ were investigated. Here, $p_{clim}$ is defined as the total number of observed rockfall events ($n_{events}$) divided by the total number of days

$(n_{\text{days}})$ in the meteorological data set:

$$p_{\text{clim}} = \frac{n_{\text{events}}}{n_{\text{days}}}. \tag{2}$$

The trend was regarded as statistically significant if a Mann-Kendall test (Wilks, 2011) indicates a significance level of less than 5%.

Next, the statistical model (Eq. 1) is applied to regional climate model data in order to investigate changes in rockfall probability under future climate scenario conditions. As variables in climate simulations can exhibit a bias, a bias correction was applied to the daily maximum and minimum temperatures before determining the freeze-thaw cycles. The correction term

was determined by evaluating the percentile that corresponds to $0°C$ at each grid point in the E-OBS data set and finding the temperature corresponding to this percentile in the regional models. Subtracting this temperature results in a quantile correction at $0°C$.

For precipitation and the fissure water proxy $D$ no bias correction is necessary as only percentiles for both parameters are used as input for the statistical model. The reference for these percentiles are the model years 1971-2000 of the historical

simulation of each respective model. In order to map $D$ and precipitation to percentiles a function is fitted that describes the probability density of $D$ and precipitation. For $D$ it was found that a good fit is achieved using a two-parameter beta function, while precipitation is better described by a gamma function. For daily precipitation local percentiles are required. The fit is therefore conducted for each grid point separately. For $D$ the fit is determined for all land grid points over the CE study region together.

In order to analyse the climate change signal, the last 30 years of the 20th century are compared to the last 30 years of the 21st century. The first period stems from the historical simulations forced with observed greenhouse gas concentrations and the second period is part of the RCP8.5 scenario simulations. The number of days favourable for rockfalls ($p>p_{\text{clim}}$) are counted for both periods, with $p_{\text{clim}}$ defined as the local average over all probabilities in the historical period.

Following Jacob et al. (2014), the multi-model result at a grid box is regarded as robust if 66% of the models agree on the

direction of the change. The significance level of the difference between the two periods is calculated for each grid box in each model simulation individually using a Monte Carlo technique. The idea of the test is to determine how likely it is to obtain the same or a greater difference when a random generator distributes the favourable days into the two periods. The implementation can be best explained with an example: Assuming the number of favourable days at one grid point in model A in the historical period is 2000 and in the scenario period it is 1500. Together there are 2500 events and the difference between the two periods

is $\Delta$=-500. Each of the 2500 events will now be individually assigned to one of the two periods by the random generator and the difference $\Delta_{random}$ between the two periods is determined. This process is repeated 100 times producing 100 values for $\Delta_{random}$. If $\Delta$ is larger or smaller than 95% of these values, $\Delta$ is regarded as statistically significant at the 5 % level. The multi-model result at a grid point is regarded as statistically significant if 66% of all models exhibit a significance level of no more than 5% and agree on the direction of the change.

## 3.2 Large-scale weather patterns

Patterns of meteorological conditions associated with rockfalls are identified by classifying the days with an event in the rockfall data base, using the objective classification algorithm SANDRA (Philipp et al., 2007). Input variables are taken from the ERA5 reanalysis data set. As input variables we tested anomalies of the mean sea level pressure (MSLP) with respect to the long term mean and anomalies of the 500 hPa geopotential height (GPH500) for two different domain sizes. The first domain covers the greater European (GE) region and the second one the smaller European (SE) domain. For SANDRA the desired number of clusters (i.e. weather patterns) needs to be specified by the user and we tested numbers between 2 and 12. The classification algorithm groups the meteorological fields at the rockfall days in such a way that the within-cluster variance of the Euclidean distance $E$ between the centroid of the cluster $\overline{X}$ and the cluster elements $X_i$ is minimized. The Euclidean distance is calculated by summing up over all grid points of the field (Eq. 3), with $g$ denoting the grid point and $ng$ being the number of points in the area of interest.

$$E(X_i, \overline{X}) = \left[ \sum_{g=1}^{ng} (X_{ig} - \overline{X_g})^2 \right]^{\frac{1}{2}},$$ (3)

By design each event day is assigned to a cluster. We are, however, only interested in groups of events that have occurred under similar large-scale weather conditions (patterns). Events that don't fit into groups need to be removed. We therefore refined the classification by testing if excluding atypical rockfall days from the classification improves the results. This was done by iteratively removing a certain percentage of cluster elements (days) from each cluster. In each iteration step the element with the largest distance $E$ was removed and a new centroid was calculated. For each cluster the maximum distance between the remaining cluster elements and the centroid was stored as a threshold distance $E_{\text{thres}}$. We removed between 10% and 50% of the cluster elements for each cluster and compared the results. The selection of the best classification was based on three different criteria:

a) high ability to discriminate between favourable and unfavourable meteorological conditions with respect to rockfalls

b) the number of rockfall days in the relevant clusters should be high

c) the centroids of the relevant clusters represent distinct meteorological conditions

To evaluate the ability of a classification to discriminate between favourable and unfavourable meteorological conditions with respect to rockfall probability, the non-rockfall days were assigned to the existing clusters if $E < E_{\text{thres}}$ or to the new group "Other". For each cluster a $X^2$ test was applied to determine if the probability for rockfall occurrence in the days belonging to the cluster differs significantly from $p_{\text{clim}}$. Only the clusters that are associated with a rockfall probability higher than $p_{\text{clim}}$, pass the $X^2$-test at the 5% significance level, and contain at least 10 rockfall days were regarded as relevant. The mean probability for rockfall in the relevant clusters was determined ($p_{\text{clusters}}$). The probability increase ($p_{\text{clusters}}$ - $p_{\text{clim}}$) was taken as a measure for quality criterium a). The number of rockfall days in the relevant classes was used as quality criterium b). Criterium c) was determined by a subjective visual inspection of the centroids. We found that for more than 4 relevant classes centroids resembled each other showing only slight shifts in the location of the low pressure systems. Therefore, classifications with more than 4 relevant classes were dismissed.

The climate change signal for large scale weather patterns for the past decades and for the 21st century was determined by assigning each day of the ERA5 reanalysis/scenario simulations to a weather pattern (cluster) if $E < E_{\text{thres}}$. The Poisson trend of the annual number of days belonging to one of the relevant weather patterns was determined. Again, a trend was regarded as statistical significant if a significance level of less than 5% was reached according to a Mann-Kendall test.

## 4 Results

### 4.1 Logistic regression

By entering meteorological observations into the logistic regression model, changes in rockfall probability caused by changes in the meteorological conditions can be studied for the past. For the period between 1950 and 2020 the trends in the annual number of days with $p > p_{\text{clim}}$ at the rockfall sites is depicted in Fig. 1. The trend at most event sites is negative (blue circles). The trend is statistically significant only at 9% of the sites (dark blue circles). The decrease at these sites is small and amounts to -1.5– -3.5% per decade relative to the mean annual frequency. Statistically significant positive trends are absent.

Using variables from climate scenario simulations as input to the logistic regression model, the possible influence of climate change on rockfalls for future periods is estimated (Fig. 2). Comparing the last 30 years of the 20th century with the last 30 years of the 21st century the multi-model ensemble suggests a decrease in the annual number of days with conditions associated with $p$ exceeding $p_{\text{clim}}$ almost everywhere in the CE region. There is an overall gradient of the strength of the signal from the north-east to the south-west of the study region with the strongest decreases reaching -20% . The average decrease over Germany is between -5% and -10%. The criterion for robustness is fulfilled for most of the CE study area. The climate change signal can be regarded as statistically significant in most areas where the climate change signal stronger than -5%. Robustness and statistical significance decrease towards the east of the study area, in high mountain areas, and with proximity to the coasts. To increase confidence in the results, the number of models that predict a statistically significant change pointing into the opposite direction to the multi-model mean was counted. In the regions showing robust and statistically significant results we detected only few outliers, with counts varying between zero and one (not shown).

Entangling the effects of the different variables on the overall climate signal shows that the strongest contribution is found for the number of freeze-thaw cycles, which is reduced in the climate scenario simulations by more than 50% (Fig. 3). The precipitation sum is predicted to increase. In the long-term mean this gain is, however, compensated by the increase in potential evapotranspiration. The number of days with precipitation exceeding the median for the historical period (using only days with precipitation) remains almost unchanged. The strongest precipitation increase is simulated by the model combination IPSL-IPSL-CM5A-MR IPSL-INERIS-WRF331F-v1. This is the only simulation predicting an area-mean increase in rockfall probability in the study region.

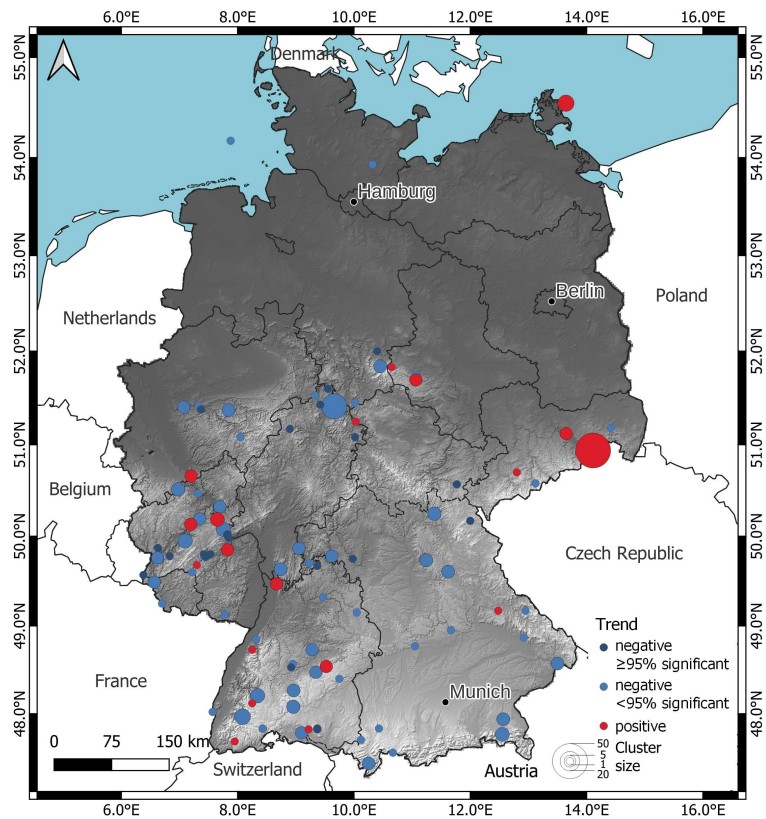

**Figure 1.** Trend in the annual number of days showing a probability for rockfalls that is higher than the climatological probability. Shown is the result at the rockfall sites determined with the logistic regression model using observations. Blue denotes a decrease and red denotes an increase. A darker colour shade is used if a Mann-Kendall test indicated statistical significance at the 5% level.

## 4.2  Large-scale weather patterns

Comparing GPH500 and MSLP anomalies as input variables for the SANDRA classification, showed that GPH500 always performed better in terms of the three quality criteria defined in Sec. 3. Using the larger GE domain generally led to better
results than the smaller SE domain. Removing the rockfall days from the analysis that are most dissimilar to the cluster centroids, reduces the number of events in the relevant clusters but increases the ability of the classes to discriminate between favourable and unfavourable conditions. A good balance was achieved by grouping GPH500 anomalies in the greater European domain in 10 clusters and afterwards removing 20% of the event days from each cluster. Four of the resulting clusters fulfil the criteria for a relevant weather pattern described in Sec. 3 (Fig. 4). 34% (88) of the events are contained in these four clusters
but the clusters occur on only 16% of all days. The probability for the occurrence of a rockfall event is therefore increased by 106% (between 78 and 193% in the individual classes).

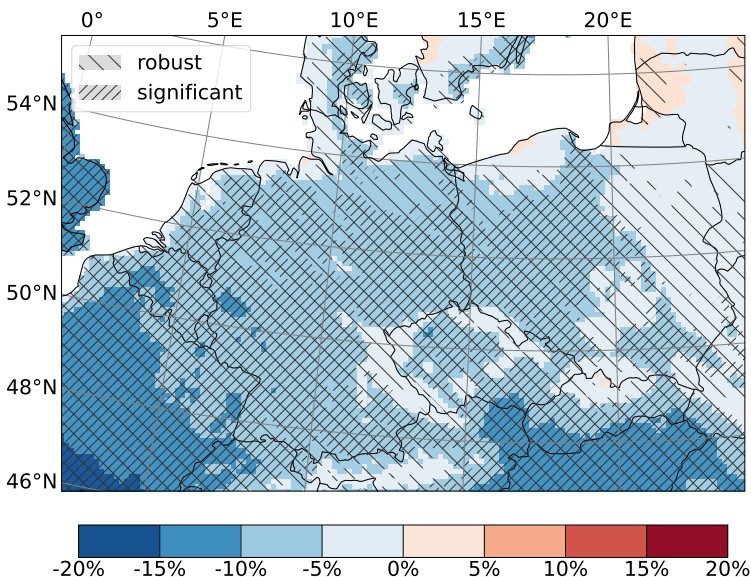

**Figure 2.** Difference in % between the simulation periods 1971-2000 and 2071-2100 for the number of days showing a probability for rockfalls that is higher than the climatological probability. Hatching denotes regions for which the signal is statistically significant and/or robust.

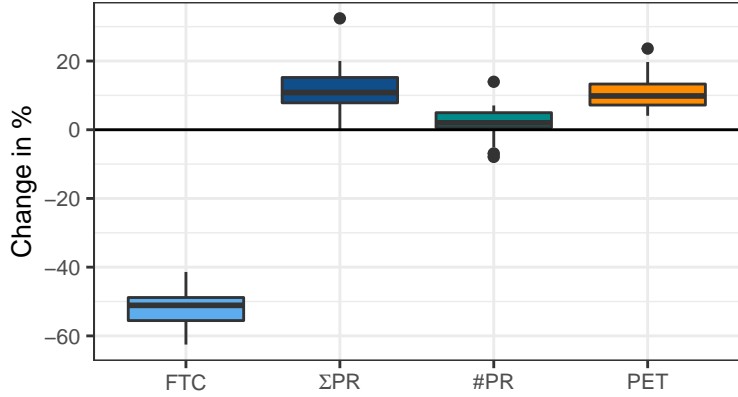

**Figure 3.** Difference in % between the simulation periods 1971-2000 and 2071-2100. The boxplot shows median and quartiles of the individual model simulations. Changes are shown for the number of transitions from freezing to thawing (FTC), the time mean of precipitation ($\sum$ PR), the number of days with precipitation higher than the median (#PR) and mean potential evapotranspiration (PET). Values are averaged over the region depicted in Fig. 2.

Composites over all days belonging to the relevant weather patterns are depicted in Fig. 5. Weather pattern 6 that exhibits the highest increase in rockfall probability is characterised by an upper air trough over Germany. Patterns 2 and 3 are associated

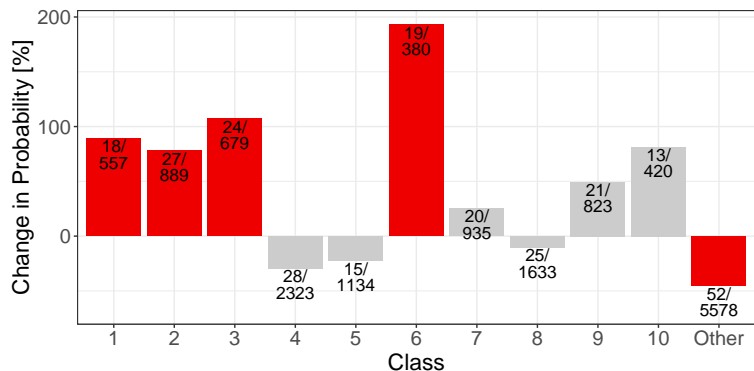

**Figure 4.** Difference in % between the climatological rockfall probability and the probability within a class. A statistically significant difference from climatology is denoted by a red bar. The number of rockfall days and the total number of days for each class are given.

with a trough over the East Atlantic/North Sea respectively while pattern 1 shows a wave structure with positive anomalies over Britain and negative anomalies over Eastern Europe. Patterns 2 and 3 are accompanied by high humidity over Germany. The seasonal cycle of the relevant weather patterns (Fig. 6) reveals that the frequency of all four patterns exhibits a minimum in summer. Pattern 1 and 2 are most frequent in April (spring), while pattern 6 shows the highest frequency in February (winter). Pattern 3 mostly occurs between November and April. Consistent with the concentration of relevant weather patterns in the cold half of the year, the number of freeze-thaw cycles associated with them is higher than climatologically expected, when taking the entire year into account (Y). Looking at individual months the number of freeze-thaw cycles doesn't show homogeneously high values for these patterns compared to other days during the cold half of the year. The relevance of patterns 1,2 and 6 is therefore mostly explainable with the fact that they are associated with winter and spring conditions. Consistent with the positive moisture anomalies found for pattern 2 and 3 (Fig. 5) these patterns are also associated with positive annual-mean precipitation anomalies (Y). This anomaly is especially strong for pattern 3 that exhibits positive precipitation deviations from November to April.

The frequency of days associated with the relevant clusters does not show a statistically significant trend for the ERA5 period as the variability in the number of days per year is too high for a significant signal (not shown). This is different when analysing the climate scenario simulations. After interpolating the simulated GPH500 anomalies in the GE region to the $1° \times 1°$ grid they were assigned to the clusters and counted. It turns out that the number of days per year in the relevant classes decreases over the scenario period in all simulations (Fig. 7). The trends for all simulations considered are statistically significant on the 5% level.

## 5 Discussion

The analysis of meteorological influences and the possible impact of climate change on rockfalls and other mass movement types are difficult to assess (Crozier, 2010; Dijkstra and Dixon, 2010; Huggel et al., 2012; Gariano and Guzzetti, 2016). Among

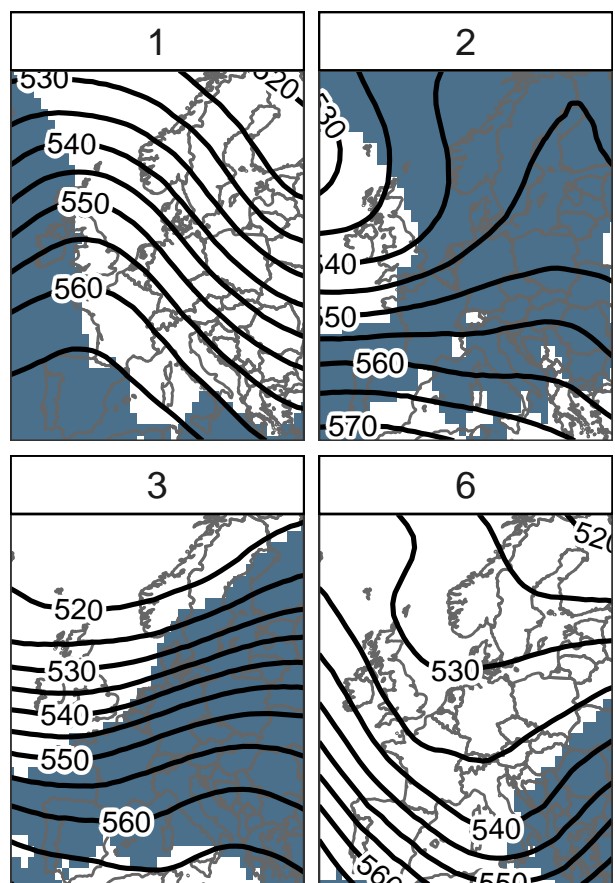

**Figure 5.** Composites of 500 hPa geopotential height in decametre (isolines) and positive monthly moisture anomalies (shading) for the 4 classes exhibiting a statistically significant potential for rockfall (1, 2, 3 and 6).

other things, this is due to the spatial-temporal inhomogeneity of rockfall datasets compared to the abundance and homogeneity of climate data. There is a general problem with historical rockfall data for which time and location need to be determined retrospectively. Moreover, supra-regional statements such as those made by the IPCC (IPCC, 2012, 2013) for an increase of rockfall probability for the whole of Central Europe, must be critically reviewed. In the context of the current climatic development in Central Europe, a general increase in process activity does not follow. Thereby, the distinction between regions with or without permafrost plays a particularly decisive role as permafrost degradation under a rise in temperature causes rock wall instability that can lead to an increase in rockfall events (Gruber and Haeberli, 2007; Paranunzio et al., 2016; Ravanel and Deline, 2011; Viani et al., 2020). In contrast, a decrease in rockfall probability needs to be considered due to a potentially balancing effect of increased evapotranspiration from increased temperatures as well as temperature driven changes in soil water regimes (Dehn and Buma, 1999; Collison et al., 2000). The statistical model applied in this study therefore considers the combined effect of precipitation, freeze-thaw cycles and fissure water (Nissen et al., 2022). The model was developed for

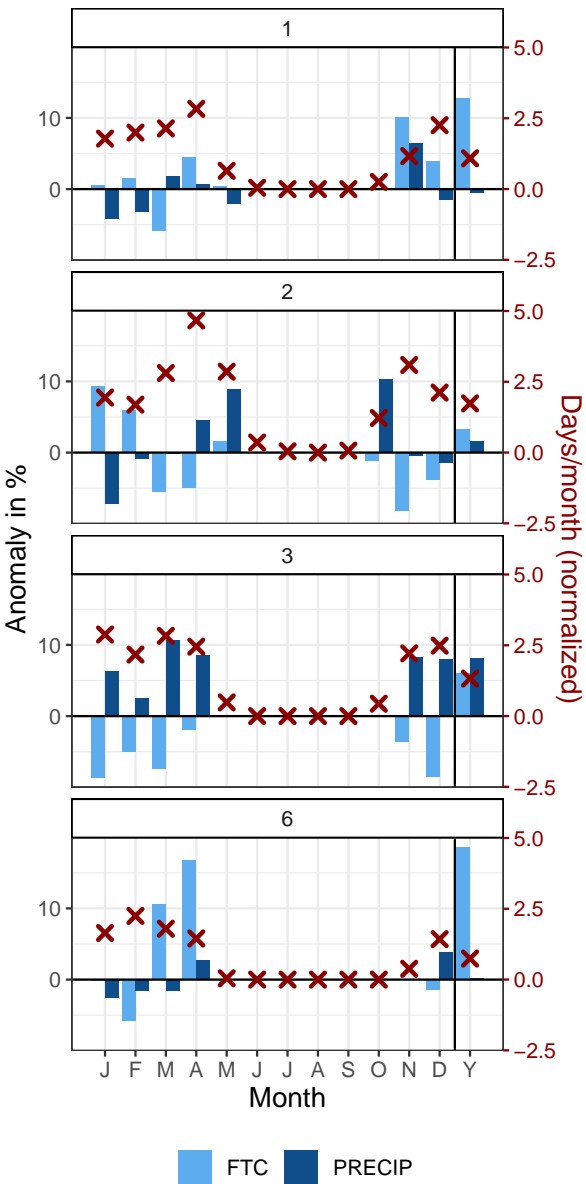

**Figure 6.** Anomalies in the meteorological conditions at the rockfall sites at the days associated with the 4 large-scale weather patterns exhibiting a statistically significant potential for rockfalls. Shown is the percentage change compared to climatological conditions for the frequency of freeze-thaw cycles (FTC, light blue) and days with precipitation above the median (PRECIP, dark blue). The comparsion is shown for the entire year (Y) and individual months. The red crosses denote the mean number of days per month for which the weather pattern was detected, normalized to a 30-day month.

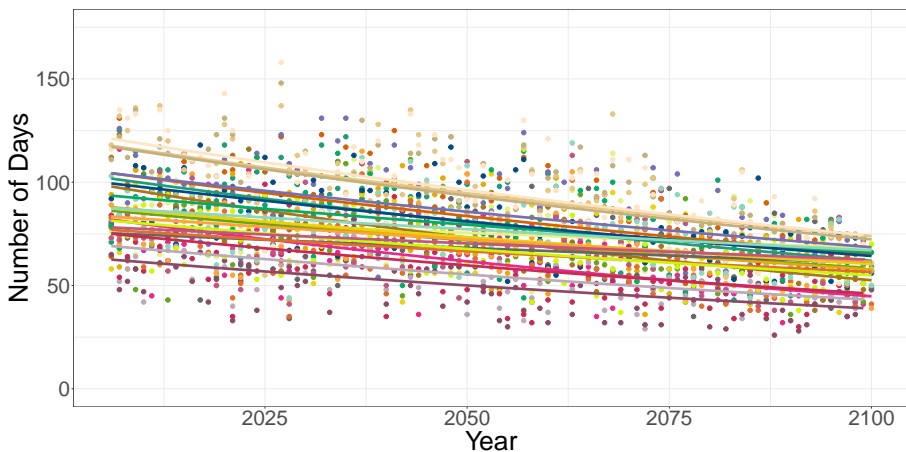

**Figure 7.** Annual number of days associated with weather classes 1,2,3 and 6 in the RCP8.5 scenario simulations. The line indicates the trend determined with Poisson regression. A solid line indicates a statistically significant trend. Each colour represents a different model.

rockfall occurrences in the German lower mountain ranges, a study region that does not include areas with permafrost. In fact, the model predicts a decrease in rockfall probability under the investigated climate scenario conditions in the German lower mountain ranges. Determining the exact region for which the statistical model can be regarded as representative was beyond the scope of this study. We analysed an area notedly larger than Germany, which illustrates that similar changes in the
meteorological conditions affect the entire study area.

In a previous study we have shown that precipitation is the most effective triggering factor for rockfalls in the German lower mountain ranges (Nissen et al., 2022). Even though heavy precipitation is predicted to increase as a consequence of global warming (e.g. Jacob et al., 2014; Nissen and Ulbrich, 2017; Seneviratne et al., 2021) this effect is counterbalanced by an increase in evaporation. The strongest factor contributing to the decrease that was found in this study is the decrease in
the number of freeze-thaw cycles associated with the overall warming trend of the atmosphere. The RCP8.5 scenario that was analysed in this study represents the upper end of the potential change and may not be the most realistic scenario (Hausfather and Peters, 2020). However, previous studies have shown that the two variables that drive the changes in rockfall probability in our study region, temperature and precipitation, change in the same direction and exhibit similar spatial patterns also in more moderate emission scenarios (Jacob et al., 2014; Nissen and Ulbrich, 2017). Future changes in rockfall probability may
therefore turn out to be weaker than suggested by our study but can be expected to show the same sign and spatial distribution.

Large-scale weather patterns associated with a high probability for rockfall initialisation occur most often in winter and spring, which is in line with the seasonal cycle of rockfalls itself (Rupp and Damm, 2020). The negative trend found with the logistic regression approach is confirmed by the decrease in the frequency of these patterns. For the identification of large-scale weather patterns, we only took the day of the event into account. Antecedent conditions, which also play a role (e.g.
for the fissure water) were neglected in the weather-pattern approach. This simplification does not affect the overall result on the direction of the trend, that is dominated by the decrease in the number of cold-season patterns that are associated with the

occurrence of freeze-thawing cycles. In combination, both approaches draw a consistent picture of the future trend in rockfall probability in the German lower mountain region under climate change conditions.

## 6    Conclusions

The application of a logistic regression model and the identification of rockfall-promoting large-scale weather patterns with input variables from climate scenario simulations enabled the analysis of rockfall probability under climate change conditions. Both methods suggest that rockfall probability decreases under RCP8.5 scenario conditions. The impact data set that was used to establish the relationship between the meteorological variables and the events, comprises rockfall data from the German lower mountain ranges and should also be applicable for further low mountain regions with similar characteristics in Central Europe.

The chapter "Climate Change Information for Regional Impact and for Risk Assessment" of the last IPCC report (Ranasinghe et al., 2021) summarizes the results of recent publications on the impact of climate change on landslides and rockfalls for various regions of the world. The review imparts the impression that climate change will increase landslide frequency almost everywhere, even though the confidence in the results is ranked as low. Considering studies on rockfalls in particular, it becomes apparent that most studies serving as a basis for the IPCC statement concentrate on high mountain areas, which are usually related to permafrost (Gruber and Haeberli, 2007; Ravanel and Deline, 2011, 2015). The few studies in lower mountain regions that analyse the influence of meteorological factors on rockfalls based on observations (Sass and Oberlechner, 2012; Mainieri et al., 2022) don't find any significant trends. This study has shown that for the subclass of rockfall events in the German low mountain ranges climate change will most likely lead to a decrease in the probability of events.

This study is the first that investigates the role of climate change on rockfalls in Germany. The analysis demonstrates the importance of considering all meteorological variables that influence rockfall probability together, as they are affected differently by climate change. The method used to describe the interplay of the different variables at different time lags is logistic regression. So far, this method has only been used for this purpose in the context of early warning systems (Abaker et al., 2021). Our study is the first to apply it to a multi-model ensemble of regional climate scenario simulations. The second method we apply in a new context is cluster analysis. The analysis is used to identify large-scale weather patterns associated with conditions that promote rockfalls. Projected changes in the frequency of these patterns are determined. Both methods proved useful to study the effect of climate change on rockfalls and have the potential to be also be applied in other regions and for other types of mass movements such as landslides.

*Code and data availability*.  The SANDRA classification scheme is part of the COST733class software package and available from COST Action 733 (2022). The EURO CORDEX simulations are available from https://esgf-data.dkrz.de (last access: 1.11.2022). The ERA5 reanalysis data set is available from Hersbach et al. (2018). The E-OBS dataset can be downloaded from https://www.ecad.eu/download/ensembles/ensembles.php (European Climate Assessment and Dataset, 2022). REGNIE is available from http://opendata.dwd.de/climate_environment/CDC/grids_germany/daily/regnie/. Information on the rockfall events can be found in the supporting material of Rupp and Damm (2020).

*Author contributions.* **Katrin M. Nissen**: Conceptualization, Methodology, Formal analysis, Investigation, Data Curation, Writing — Original Draft, Writing — Review & Editing, Visualization. **Martina Wilde**: Investigation, Writing — Original Draft, Writing — Review & Editing. **Thomas Kreuzer**: Investigation, Writing — Original Draft, Writing — Review & Editing. **Annika Wohlers**: Writing — Original Draft, Visualization. **Bodo Damm**: Investigation, Resources, Writing — Review & Editing, Supervision, Project administration, Funding acquisition. **Uwe Ulbrich**: Writing — Review & Editing, Supervision, Project administration, Funding acquisition.

*Competing interests.* One of the (co-)authors is a member of the editorial board of Natural Hazards and Earth System Sciences but was not involved in the editorial or review process regarding the present paper.

*Acknowledgements.* This study was funded by the Federal Ministry of Education and Research in Germany (BMBF) through the research program ClimXtreme (FKZ: 01LP1903A, 01LP1903K) The work used resources of the Deutsches Klimarechenzentrum (DKRZ) granted by its Scientific Steering Committee (WLA) under project IDs b1152 and bm1159. We acknowledge the ERA5, E-OBS, REGNIE, and EURO CORDEX datasets. We thank the two anonymous reviewers and our editor for their constructive comments.

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



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
