# Peer review of "A decrease in rockfall probability under climate change conditions in Germany"

_EGUsphere, 2023_

## Author Comment (AC1)

Fig: Number of models that disagree with the multi-model mean shown in Fig. 2. For these models the difference in the number of events between the historical and the scenario period is statistically significant at the 95% level but of the opposite sign.

---

## Author Response (AR1)

**Point-by-point reply**

**Review 1:**

- Line 39. Some interesting references: https://doi.org/10.1007/s11069-017-3003-3 ; https://doi.org/10.5194/nhess-16-2085-2016 ; https://doi.org/10.1016/j.scitotenv.2014.02.102.

*Thanks for pointing these out. We included them in the revised version of the manuscript.*

- Line 60. Do you have any information on the accuracy with which the rockfalls were mapped or localized spatially?

*For the events from the landslide database for Germany the precise location is recorded. We added this to the manuscript. The potential deficits of the railway data set were already part of the manuscript.*

- Line 74. Any consideration regarding the use of rainfall data with a daily temporal resolution for rockfalls?

*In Nissen et al. 2022 we compared hourly rainfall and daily rainfall as predictors for the logistic regression model. It turned out that daily precipitation lead to a better fit of the model. This is not discussed again in this paper.*

- Line 76. Are the E-OBS and REGNIE spatial resolutions in accordance with the spatial resolution of the rockfall database? Or perhaps they are too coarse?

*Both data sets are constructed by interpolating station data to a regular grid considering orographical conditions. The value in a grid box represents the mean over the area covered by the grid box. Its accuracy depends for example on the station density in the vicinity and the spatial homogeneity of the variable. It will deviate to some extent from the exact conditions. As the exact values at the rockfall sites are not known we consider these datasets the best option we have for this kind of analysis. The results from the logistic regression model (Nissen et al. 2022) confirm that the datasets are useful for our purpose. We added a sentence explaining the spatial interpolation.*

- Line 85. Why did you decide to use the 1971-2000 period and not the "common" 30-year period 1981-2010?

*The EURO-CORDEX historical period ends in 2005.*

- Line 88. This is interesting about RCP8.5: https://www.nature.com/articles/d41586-020-00177-3. However, why have you not used also another RCP?

*As we have stated in the manuscript the idea was to capture the upper range of the potential changes. In our study area the effect of climate change on the two meteorological variables that drive the changes in rockfall probability - (extreme) precipitation and temperature - point into the same direction for the RCP4.5 and the RCP8.5 scenario (https://doi.org/10.1007/s10113-013-0499-2 and https://doi.org/10.5194/nhess-17-1177-2017). Changes are just less pronounced in RCP4.5 compared to RCP8.5. We therefore expect that applying the statistical model to RCP4.5 simulations would also show a reduction in rockfall probability with a similar spatial distribution than the one shown in our manuscript, just weaker and with lower statistical significance. We added a statement to the discussion section.*

3. Methods

- Have you checked the presence of change points and structural breaks rockfall time series (e.g. Pettitt test)?

*The rockfall time series is shown in Nissen et al. 2022 and described there in more detail. An important fact is that the rockfall database is not comprehensive. It shows an increase in the number of recorded events with time that is not due to climatic conditions but reflects the fact that data on rockfall events were more readily available in recent years. Structural breaks are therefore present. We added a sentence to our manuscript.*

- Have you checked if the series can be correlated (e.g. if a correlation coefficient can be calculated or if a Kendall test (correlation rank) can be applied)?

*As stated before the time series is not comprehensive. We have therefore not attempted any correlation analysis. In Nissen et al. 2022 we show that a relationship exists between the meteorological predictor variables and rockfall probability using Weights of Evidence.*

- Line 103. Why 9 days? Probably it was defined in Nissen et al 2022, however, I would suggest adding some details here.

*The 9-day period is a result of Nissen et al. 2022. Slightly longer or shorter periods give very similar results but the best model fit for the logistic regression model was achieved using 9 days. We added a sentence to the manuscript to clarify this.*

- Line 111. The observational period is 1950-2020, while the period considered for the present-day greenhouse as forcing is 1971-2000. Some details should be added regarding this inhomogeneity.

*For the observational period we analysed trends which requires a long continuos time series. For the greenhouse gas experiments we compare two time slices that need to be long enough to account for natural variability but short enough to be able to neglect trends. Both methods are common practice.*

- Lines 162-169. These sentences are not very clear.

*We rewrote this section.*

- Line 179. "The trend is statistically significant only at few sites". It seems they are very few. How many? What's the percentage with respect of the total?

*Only 9% (the information was added to the manuscript)*

6. Conclusions

I would suggest adding in the conclusions section more findings and the main novelty of the present work. Eventually also limitations (also considering my comments above) and future perspectives could be added.

*We extended the conclusions.*

Technical corrections

- Use either "rockfalls" or "rock falls" to be consistent in the whole text.

*We homogenized the term.*

- Line 1. change "rockfalls" or "rockfall probability"

*"s" was added*

- Line 116. perhaps "statistically significant"?

*This was corrected.*

- Figures 1 and 2. I'd suggest adding in the caption the projection used.

*As Fig 2 is not in a commonly known projection but in the model's original rotated grid. We think that adding information on projections will lead to unnecessary confusion.*

- Figure 3. Check the abbreviations in the caption and in the figure legend.

*This was corrected.*

- Figure 4. I would separate the two labels in each bar vertically, for better readability.

*The figure has been improved.*

- Figure 5. I would reduce the font of the 1,2,3,6, labels. Perhaps the a), b), c), and d) labels should be added.

*We reduced the font.*

- Figure 6 I would add the abbreviations in the caption, e.g. (FTC, light blue) and (PRECIP, dark blue).

*This was added.*

**Review 2:**

Line 75: The given values are not the horizontal resolution, but the grid spacing of the interpolation method. Actual resolution of processes may be very different from this. I cannot judge well whether this has an effect on the study here.

*This is correct. As for both variables the interpolation considers elevation, we think that using the gridded data provides more reliable estimates for the event location than using station data from the closest station. We rephrased the description.*

Line 90: Instead of choosing the first ensemble member, wouldn't it be interesting to run the method several times with changing ensemble members? This would give an indication of the stability of the solution.

*There are only 4 combinations (out of 23) of regional and global models for which more than one ensemble simulation was available. The forcing is limited to two different global models (out of 8). We think that the gain in confidence that can be achieved by sampling these 4 simulations in the multi-model ensemble is small and not representative. We have therefore decided to apply the multi-model ensemble approach that is state-of-the-art in most IPCC studies and refrain from this extra analysis.*

Line 103: Why exactly are nine days selected for the freeze-thaw cycles? How does this fit with the patterns for just one day below? The issue has been addressed, but perhaps some additional comments could be added here, with a look ahead to further work if necessary.

*The 9-day period is a result of Nissen et al. 2022. Slightly longer or shorter periods give very similar results but the best model fit for the logistic regression model was achieved using 9 days. We added a sentence to the manuscript to clarify this.*

Line 135: That only two-thirds of the models point in the right direction, at all, seems like a weak signal at first glance. Perhaps more could be explained about this. The question is also how many point significantly in the other direction (if relevant, see line 180).

*We like the suggestion on how to check the model disagreement. In the areas showing a robust and statistically significant signal in Fig. 2 the numbers are very low (0-1 models). We added a sentence to the manuscript.*

Line 136: The Monte Carlo method could be explained in more detail. At this point, it would also be interesting to see what the autocorellation of the time series looks like, which may have an influence on the assessment of statistical significance.

*There will clearly be some autocorrelation as our statistical model includes preconditions calculated over a period of time (freeze-thaw cycles in the previous 9 days and moisture preconditions calculated from the conditions in the previous 5 days). We have chosen to test for significance using the Monte Carlo technique as this method can be applied regardless of the data distribution. We have extended the explanation on the method in the manuscript using an example.*

Line 152: When fitting a statistical model, how is it justified to take out the data that does not fit the model? Maybe I have overlooked something here, but it seems like circular reasoning to me.

*We don't aim at fitting a statistical model. Instead we investigate if there are groups of events that have occurred under similar large-scale weather conditions (patterns). Events that don't fit into groups are removed. We rephrased the text to make this clearer.*

Line 180: Just for clarification, so there are no statistically significant positive trends or are they just not shown?

*As stated, there are no statistically significant positive trends for the period 1950-2020.*

Technical corrections:

Line 18: "as landslide are/is"?

*"s" was added.*

Line 151: Maybe describe g and ng briefly.

*An explanation was added.*

Figure 5: Legend for moisture field is missing.

*The explanation is given in the caption. As we only distinguish between positive and negative anomalies, we think that the legend is expendable in favour for a larger map.*

All illustrations: Font sizes and line widths vary widely between figures. Perhaps it is possible to harmonize this a little.

*We have adjusted font sizes*